# Functional Segments on Intrinsically Disordered Regions in Disease-Related Proteins

**DOI:** 10.3390/biom9030088

**Published:** 2019-03-05

**Authors:** Hiroto Anbo, Masaya Sato, Atsushi Okoshi, Satoshi Fukuchi

**Affiliations:** Department of Life Science and Informatics, Faculty of Engineering, Maebashi Institute of Technology, 460-1, Kamisadori, Maebashi, Gunma 371-0816, Japan; koume8@icloud.com (H.A.); mail_address107@icloud.com (M.S.); m1461008@maebashi-it.ac.jp (A.O.)

**Keywords:** intrinsically disordered regions, functional segments, disease-related proteins, protein-protein interaction, subcellular location

## Abstract

One of the unique characteristics of intrinsically disordered proteins (IPDs) is the existence of functional segments in intrinsically disordered regions (IDRs). A typical function of these segments is binding to partner molecules, such as proteins and DNAs. These segments play important roles in signaling pathways and transcriptional regulation. We conducted bioinformatics analysis to search these functional segments based on IDR predictions and database annotations. We found more than a thousand potential functional IDR segments in disease-related proteins. Large fractions of proteins related to cancers, congenital disorders, digestive system diseases, and reproductive system diseases have these functional IDRs. Some proteins in nervous system diseases have long functional segments in IDRs. The detailed analysis of some of these regions showed that the functional segments are located on experimentally verified IDRs. The proteins with functional IDR segments generally tend to come and go between the cytoplasm and the nucleus. Proteins involved in multiple diseases tend to have more protein-protein interactors, suggesting that hub proteins in the protein-protein interaction networks can have multiple impacts on human diseases.

## 1. Introduction

Intrinsically disordered proteins (IDPs) are proteins that do not adopt unique three-dimensional structures under physiological conditions [1,2,3]. They are fully or partially disordered and are abundant among eukaryotic proteins [4,5,6]. One of the unique features of IDPs is their ability to bind to binding partners. The regions performing such binding are generally short segments ranging from several residues to tens of residues and can adopt local two-dimensional structures in association with this binding. This has been referred to as the coupled folding and binding mechanism. These interactions are transient, specific, and low-affinity. Through this mechanism, intrinsically disordered regions (IDRs) play crucial roles in many biological processes, such as signal transduction and transcriptional regulation [1,2,3,7]. 

The importance of IDPs in human diseases has been reported [8,9]. Intrinsically disordered proteins are found in high concentrations in plaques and brain deposits in neurodegenerative patients, and mutations in IDRs can increase aggregation propensity. Intrinsically disordered proteins such as α-synuclein, the amyloid β peptide, and huntingtin have been directly linked to diseases such as Alzheimer’s, Parkinson’s, and Huntington’s diseases [10,11,12,13,14,15,16]. It has been shown that many IDPs participate in cell signaling and cancer-associated proteins [7]. Breast cancer type 1 susceptibility protein (BRCA1) is one of the most typical IDPs, with a long central region of 1480 residues shown to be disordered by nuclear magnetic resonance (NMR) and circular dichroism (CD) spectroscopy [17]. This long IDR has many binding segments for proteins such as p53, retinoblastoma protein, and the oncogenes c-Myc and JunB. p53 Is a transcription factor that has IDRs in its N- and C-terminus. These IDRs have binding sites for many partner proteins. Among these, the interaction between p53 and E3 ubiquitin-protein ligase Mdm2 (MDM2) has been given much attention in cancer research, as p53 can induce apoptosis to suppress tumor progression [18,19]. Bioinformatics work has also shown that IDRs are rich in proteins involved in cancer, neurodegenerative diseases, cardiovascular diseases, and diabetes [9,20]. Intrinsically disordered proteins have gained attention as drug targets. Inhibitors targeting IDR and globular domain interactions have been developed for the interaction between Bcl-xL and BAK [21,22], MDM2 and p53 [23], interleukin (IL)-2 receptor α and IL-2 [24,25], XIAP and Smac [26,27], and CBP and β-catenin [28]. 

As shown above, protein–protein interactions (PPIs) occurring on IDRs have high potential as drug targets. The IDP databases, IDEAL [29,30] and DisProt [31], have 913 and 803 proteins, and IDEAL has collected 559 protein-binding segments on IDRs called “protean segments (ProSs)” in the database. Protean segments are defined as sequences with experimental evidence of being both disordered in an isolated state and ordered in a binding state. In contrast, several tools for predicting such binding regions have been developed to suggest that there are more than 100,000 protein-binding segments in the human proteome [32]. Considering this prediction, our knowledge on IDR-mediated interactions is still limited because the number of ProSs with experimental evidence of ordered and disordered states is only about 600. However, we have a lot of PPI data accumulating and several computer programs to predict IDRs. The performance of IDR predictions has reached a standard for practical use, and PPI annotations found in predicted IDRs can be considered protein-binding segments in IDRs. In this study, we combined the annotations of the UniProt database and IDR predictions to find these possible protein-binding regions on IDRs and analyzed these regions in the context of human disease.

## 2. Materials and Methods 

We selected human proteins from the Swiss-Prot section of the UniProt XML file [33] from UniProt release 2018_07. We extracted the feature (FT) section information. A single FT section has a feature type, a description, and a location, and all of them were extracted. The IDEAL database provides binding segments in IDRs as “ProSs”. Protean segments are mostly short segments consisting of less than 30 amino acid residues to which more than 80% of ProSs belong. Thus, we selected for feature information shorter than 30 residues. Next, we picked binding-associated features from the selected features. By manually surveying feature descriptions, the features “region of interest” and “mutagenesis site” are found to contain binding-related features. Adding to these two features, “short sequence motif” also contains functional segments in IDRs. Out of the selected features of “region of interest”, the features having the terms “interact”, “bind”, or “motif” in their description were selected. Of the “mutagenesis site” features, those with “interact” or “bind” were selected. From the selected features of “mutagenesis site”, those having “no” or “not” were discarded. We wrote some in-house scripts to find the selected features located in IDRs. A feature region found in an IDR was defined as a possible ProS, hereafter referred to as a pProS. 

We used three predictors for IDR prediction. MobiDB [31] provided predicted IDRs for several proteome datasets. We downloaded the human proteome dataset and used MobiDB-lite [34] predictions for predicted IDRs. MobiDB-lite uses eight different predictors, three variants of ESpritz (DisProt, NMR, X-ray), two variants of IUpred (long, short), two variants of DisEMBL (465, hot loops), and GlobPlot. MobiDB-lite uses the results of these predictors to combine them into a consensus result, where at least five out of eight methods must define a residue as disordered. Thus, the predicted IDRs reflect different types of predictions at one time. DISOPRED3 [35] is an extension of the previous program, DISOPRED2, to improve predictions of long IDRs. In order to achieve this goal, DISOPRED3 uses a neural network-based predictors trained on a dataset rich in long IDRs. The training was done on a position-specific scoring matrix (PSSM) generated by PSI-BLAST. This program is one of the benchmark IDR predictors, which was one of the top ranked predictors in CASP10 [36]. DICHOT is the predictor combining the homology-based domain assignments and the support vector machine learning. First, it conducts a PSI-BLAST search against the Protein Data Bank (PDB) to mask structural regions, and then the unmasked regions are judged by using the support vector machine-based predictor trained on a multiple alignment of homologs. This predictor divides an entire amino acid chain into structural domains and IDRs, which is unique compared to other predictors. We selected regions where any of the two predictors predicted an IDR.

Human disease information was obtained from the KEGG database [37]. KEGG has a collection of disease entries called KEGG DISEASE. KEGG DISEASE provides Search Disease, which is a mapping tool against disease genes accumulated in KEGG DISEASE entries. Thus, the Search Disease tool provides information on which disease a protein is involved in. The mapped disease-related proteins were divided into pProS-containing proteins and non-pProS proteins, according to the existence or absence of pProSs.

We used the UniProt annotations of subcellular locations for protein localizations. We counted terms that appeared in the annotations by the disease-related proteins and all human proteins. We defined those proteins as only having the annotation of nucleus as nuclear protein, those proteins only having that of cytoplasm as cytoplasm protein, those only having that of membrane as membrane protein, and those having both of the annotations of cytoplasm and nucleus as cytoplasm and nuclear protein. The annotations of other locations were discarded because of shortages of appearance. The disease-related proteins were further divided into pProS-containing proteins and non-pProS proteins, and the ratios of each of the terms were obtained for the disease-related proteins and all human proteins. The logarithms of the ratios of two ratios were used in analysis.

The brief outline of the procedure can be found in Appendix A.

## 3. Results

The UniProt database has 20,410 human proteins, and 3378 proteins (16.6%) were assigned human diseases by the Search Disease tool, as shown in Table 1. In the human UniProt and the disease-related proteins, 29,145 and 18,450 regions of the feature annotation shorter than 30 residues were found, respectively. From these regions, we selected the feature annotations of “region of interest”, “mutagenesis site”, and “short sequence motif” to pick up pProSs. Out of 3378 disease related proteins, 402 proteins (11.9%) had pProSs. Out of 18,450 feature annotations found in the disease related proteins, 8.3%, 3.4%, and 24.4% were found in the predicted IDRs for “region of interest”, “mutagenesis site”, and “short sequence motif”, respectively. Thus, the regions of these annotations are defined as pProSs in this study.

We illustrate how these annotations occur in IDRs in Figure 1. In this study, some of the proteins are stored in the IDEAL database [38]. In such cases, we denoted IDEAL identifiers for reference. p53 is one of the typical IDPs and has relatively short IDRs in the N- and C-terminus in addition to between the DNA-binding domain and the tetramerization domain (IDEAL: IID00015). p53 has three annotations from residues 15 to 25, one annotation from residues 48 to 56, one annotation from 305 to 321, four annotations from 359 to 363, and one annotation from 370 to 372 and 368 to 387. Among them, “TAD I”, “TAD II”, “bipartite nuclear localization signal” and “[KR]-[STA]-K motif” are from “short sequence motif”, “interaction with USP7” and “basic” are from “region of interest”, and “loss of interactions with MDM2”, “loss of interaction with PPP2R5C, PPP2CA and PPP2R1A”, and “abolishes binding to USP7” are from “mutagenesis site”. As shown in Figure 1, some of these annotations overlap each other. We counted these overlapped annotations separately in the statistics in this study because they can be different annotations even if their regions are overlapped. For example, although “TAD I” overlaps “loss of interaction with MDM2” and “loss of interaction with PPP2R5C, PPP2CA and PPP2R1A”, they describe different phenomena. Thus, we would like readers to note that the statistics in this study contain such multiple counts in some cases. 

Most of the annotations we picked in this analysis occur at one time in the dataset. As seen in the example of p53, the annotations of interest in this study are in forms such as “interaction with protein A” or “loss of interaction with protein B”, etc., which can be assigned on a single or a small numbers of proteins binding onto a specific protein. However, some of the annotations appeared multiple times. These frequently appearing annotations in pProSs are listed in Appendix A. They contain targeting sequences, such as a nuclear localization signal (NLS). Retinoblastoma-associated protein has an NLS from residues 858 to 881. This region has been described as disordered in the isolated state [39], though the binding structure with importin has been solved (IDEAL: IID00017) [40]. Another class of frequently found annotations is segments binding upon promiscuously interacting domains, such as the SH3 domain and the PDZ domain. The SH3 domain mediates PPIs via a short ambiguous peptide motif to assemble cell regulatory systems [41] and is annotated as ProS in the IDEAL database (IDEAL: IIDE00256). The PDZ domain is a scaffold protein that forms protein complexes in signaling pathways or cell trafficking [42]. The binding segment is disordered in an isolated state (IDEAL: IID9005) [43]. We also frequently found the LXXLL motif, which exists in co-repressors or co-activators of nuclear receptors. The LXXLL motif in peroxisome proliferator-activated receptor gamma co-activator 1 α has been reported as disordered, and the binding structure with steroid hormone receptor has been elucidated (IDEAL: IID00103) [44,45]. 

The disease-related proteins are classified into 15 categories by KEGG DISEASE, which are: cancers (Can); cardiovascular diseases (Car); congenital disorders of metabolism (Dme); congenital malformations (Mal); digestive system diseases (Dig); endocrine and metabolic diseases (End); immune system diseases (Imm); musculoskeletal diseases (Mus); nervous system diseases (Ner); other congenital disorders (Oco); reproductive system diseases (Rep); respiratory diseases (Res); skin diseases (Ski); urinary system diseases (Uri); and other diseases (Oth). These categories have subcategories into which these diseases are classified. The details of the disease classifications can be found on the web page for KEGG DISEASE [46].

We classified disease-related proteins into the disease categories and showed the statistics in Table 2. The rank of the numbers of unique pProSs is as follows, in descending order: “congenital malformations”; “nervous system diseases”; “cancers”; and “cardiovascular diseases”. The numbers depend on the number of proteins in each of the categories, as seen in Table 2. However, the category possessing many proteins does not necessarily contain many pProSs. For example, the “congenital disorders of metabolism” contains 687 proteins, though only 40 pProSs were identified. The values of protein coverage represent the ratio of the numbers of pProS-containing proteins to the numbers of proteins in a category. “cancers” shows the top coverage, followed by “congenital disorders,” “digestive system diseases,” and “reproductive system diseases.” The average values of annotations represent the values of the number of pProSs in a category divided by the number of pProS-containing proteins, wherein most of the values are greater than 2. As shown in Figure 1, some annotations are found in different regions, and some annotations overlap each other. The average annotation values here contain both cases. The annotations of “mutagenesis site” tend to overlap with other annotations, where 70% of them coincide with other annotations. This overlap of “mutagenesis site” may enlarge the average annotation values. The total amount of proteins with pProS over all disease categories differs from the number of pProS-containing proteins shown in Table 1. This is because some proteins were assigned to multiple diseases.

Figure 2 shows the IDR ratios by the disease categories. The IDR ratio of all disease proteins (27.0%) is similar to that of the UniProt human proteins (28.7%, the dashed line in Figure 2). However, the IDR fractions of each category differ, where “cancers,” “congenital malformations”, “other congenital disorders”, “reproductive system diseases”, and “other diseases” are over-represented. These IDR-rich categories also have high fractions of IDRs in the pProS-containing proteins. Although the IDR fractions of pProS-containing proteins correlate with the protein coverage (Table 2), a high fraction of IDRs does not necessarily mean high protein coverage. For example, although “urinary diseases” shows a very high IDR fraction in their pProS-containing proteins, the protein coverage is not high. This means that the pProS-containing proteins in this category contain high fractions of IDR, but a number of non-pProS-containing proteins also belong to this category. On the other hand, “musculoskeletal diseases” has low IDR fractions in both pProS-containing proteins and all proteins. However, the protein coverage is relatively high, suggesting that many proteins with relatively short IDR fractions have functional regions in such IDRs. Iakoucheva et al. [7] showed that cancer related proteins have high fractions of IDRs, and the result of the present analysis confirms this trend. Moreover, the current results suggest that cancer-related proteins have a lot of pProSs compared to the proteins in the other categories.

Table 3 shows the top-ranked proteins in terms of the numbers of pProS residues. Although we counted overlapped annotations separately in Table 2, we counted “pPros residues” without redundancy. For example, when there are annotations of “from 360 to 365” and “from 360 to 362”, we counted six for this region. “Cancers” and p53 appear again, and the proteins belonging to “nervous system diseases” are listed frequently. As shown in Figure 2, “cancers” has the largest IDR fraction and the top ranked protein coverage. Thus, it is natural for the proteins in “cancers” to contain proteins with a large fraction of pProS residues. On the other hand, “nervous system diseases” has an IDR fraction smaller than that of the UniProt human proteins, and the protein coverage is low. Therefore, the proteins in “nervous system diseases” do not generally contain a large fraction of IDR and pProS. However, some of them have long pProSs, even in relatively short IDRs. On the other hand, “nervous system diseases” has an IDR fraction smaller than that of the UniProt human proteins. Raychaudhuri et al. reported similar results, where in the nervous system disease related proteins, Huntington’s and Alzheimer’s disease proteins have large fractions of IDRs, and Parkinson’s disease proteins do not [20], suggesting the similarity of the average IDR fraction of these nervous disease proteins to that of UniProt human proteins. Therefore, the proteins in “nervous system diseases” do not generally contain a large fraction of IDR and pProS. However, some of them have long pProSs, even in relatively short IDRs. The example of “nervous system diseases” suggests that even if the IDR fraction and protein coverage of pProS-containing proteins are low, some proteins in a disease category can have considerable lengths of pProSs. 

In Figure 3 and the following sections, we illustrate how annotations can be found in IDRs of the disease-related proteins.

### 3.1. Eukaryotic Translation Initiation Factor 4 Gamma 1 

Parkinson’s disease is a progressive neurodegenerative movement disorder caused by the death of dopaminergic neurons in the substantia nigra pars compacta (KEGG: H00057). Although deleterious mutations in α-synuclein (OMIM-163890) [47], leucine-rich repeat kinase 2 (OMIM-609007) [48], vesicular protein sorting 35 (OMIM-601501) [49], parkin (OMIM-602544) [50], PTEN induced putative kinase 1 (OMIM-608309) [51], and DJ-1 (OMIM-602533) [52] have been found in multi-incident families with parkinsonism, mutations in the translation initiator, eukaryotic translation initiation factor 4 gamma 1 (eIF4G1, UniProt: Q04637), have also been reported [53]. The eIF4G1 is a component of the protein complex eIF4F, which is involved in the recognition of the mRNA cap, ATP-dependent unwinding of the 5’-terminal secondary structure and recruitment of mRNA to the ribosome. eIF4G1 has long IDRs in its 1599 amino acid residues. In the IDRs, the region from residues 172 to 200 possesses polyadenylate-binding protein 1 (PABPC) binding ability [54], and the region from residues 1585 to 1599 has an annotation of “necessary for binding of MAP kinase-interacting serine/threonine-protein kinase 1 (MKNK1)”. Although IDRs in this study were defined by computer predictions, the region of PABPC binding is reported to be unfolded in the isolated state [54]. The eIF4G1 has been found to have five Parkinson’s disease-associated mutations: Ala502Val; Gly686Cys; Ser1164Arg; Arg1197Trp; and Arg1205His. Out of five mutations, Ala502Val and Arg1205His appear to disrupt eIF4E or eIF3E binding and share haplotypes consistent with ancestral founders [53]. These two mutations reside on IDRs, suggesting an association of binding sites on IDRs with Parkinson’s disease.

### 3.2. Survival of Motor Neuron Protein

Spinal muscular atrophy (SMA) is a neuromuscular disease characterized by degeneration of motor neurons, resulting in progressive muscle atrophy and paralysis. The most common form of SMA is caused by mutation of the survival of motor neuron (SMN, UniProt: Q16637) protein (KEGG: H00455). Survival of motor neuron (SMN) protein is in a complex with several proteins, including Gemin2, Gemin3, and Gemin4, and plays important roles in small nuclear ribonucleoprotein (snRNP) biogenesis and pre-mRNA splicing. The SMN protein has two highly homologous genes, SMN1 and SMN2, which lie within the telomeric and centromeric halves of a large inverted repeat on chromosome 5q. The coding sequence of SMN2 differs from that of SMN1 by a single nucleotide in exon 7 (840C-T) [55,56,57]. Thirty-eight patients with SMA have a homozygous deletion of exon 7 of the SMN1 gene, and the deletion is associated with homozygous deletion of exon 8 in 31 of 34 patients [58]. Exon 7 and 8 cover the region from residues 242 to 294, which is predicted to be an IDR. The region from residues 252 to 281 contains the YG-box for forming helical oligomers (PDB: 4gli) [59]. In the oligomers, the C-terminal region from residues 281 to 297 is disordered [59]. This C-terminal region contains a binding region for heterogeneous nuclear ribonucleoprotein (hnRNP) Q, and the most common SMN mutant found in SMA patients is defective in its interactions with snRNPs [60]. Thus, SMN protein provides an example that a defect of IDR binding ability causes a serious disorder. SMN has another IDR in its N-terminal; this region also contains binding sites for GEMIN2, though the length of the annotation and IDR coverage do not meet the threshold of pProS in this study. 

### 3.3. Low-Density Lipoprotein Receptor Adaptor Protein 1 

A known cause of hypercholesterolemia is deficiency of low-density lipoprotein receptors (LDLR) or apolipoprotein B [61]. Recently, mutations in the low-density lipoprotein receptor adapter protein 1 (ARH, UniProt: Q5SW96) were found to cause the autosomal recessive form of hypercholesterolemia [62]. Low-density lipoprotein receptors -mediated endocytosis in the liver is the primary pathway for clearance of circulating LDL, to prevent LDL accumulation. The ARH protein is an adaptor protein required for efficient endocytosis of the LDL receptor. The ARH protein can interact with the internalization sequence in the cytoplasmic tail of LDLR, and the N-terminal region of the clathrin heavy chain, a component of a polyhedral lattice on the transport vesicles in the clathrin-mediated membrane traffic. It also binds upon the beta subunit of adaptor protein complex 2 (AP-2), which is a vesicle coat component involved in cargo selection and vesicle formation. The ARH protein is predicted to have two IDRs in its N- and C-terminus. The IDRs in the C-terminal end have two functional regions: clathrin binding and AP-2 complex binding, which are crucial interactions for LDLR-mediated endocytosis. Although the IDRs are defined only by the computer predictions, the clathrin-binding region of auxilin has been found to be disordered [63], and the tertiary structure of the AP-2 complex binding region of ARH ([DE]-X(1,2)-F-X-X-[FL]-X-X-X-R motif) in the complex with AP-2 β subunit has been solved (PDB: 2g30) [64]. Appendix A shows the structure of this segment binding upon AP-2 β subunit, together with a typical ProS structure of nuclear receptor co-activator 1 (IDEAL: IID50084) binding upon nuclear receptor subfamily 1 group I. Nuclear receptor co-activators or co-repressors have the LXXLL motif in their binding site for nuclear receptors. This motif is one of the annotations frequently found to be a pProS in this study (Appendix A) and has been shown to be disordered in the isolated state, as mentioned above. The AP-2 binding region shows a similar α-helix structure attached to the globular structure of the partner protein.

### 3.4. Desmin

Desminopathy belongs to a genetically heterogeneous group of disorders named myofibrillar myopathy, caused by mutations in desmin, αB-crystallin, myotilin, Z-band alternatively spliced PDZ-containing protein, filamin, or Bcl-2-associated athanogene. Desmin (Uniprot: P17661) is the main intermediate filament (IF) protein. It interacts with other proteins to support myofibrils at the level of the Z-disc and forms a continuous cytoskeletal IF network [65]. Desmin has two predicted IDRs in its N- and C-terminus, and the binding region for αB-crystallin (CRYAB) is located in the C-terminal IDR [66]. Several mutations in this region have been reported to cause severe disturbance of filament formation. The desmin mutations Thr442Ile, Arg454Trp, and Ser460Ile reveal a severe disturbance of filament formation competence and filament interactions [66,67,68], and Thr453Ile exhibits significantly delayed filament assembly kinetics [69,70]. These sites locate on one of the predicted IDRs.

We analyzed the disease-related proteins from subcellular localizations in Figure 4. The location distributions of all disease-related proteins (“all” shown in the right bottom panel) showed that pProS-containing proteins are rich in cytoplasm and nuclear (CN) proteins and nucleus (N) proteins. The distribution of the non-pProS proteins shows a shortage of CN and N proteins. Ota et al. [71] pointed out the IDR richness in the mobile proteins shuttling from cytoplasm to nucleus and vice versa. Cytoplasm and nuclear proteins localize both in the cytoplasm and nucleus, and thus, the results of this study also showed that these CN proteins have functional regions on their IDRs. Although most of the localizations by the disease category showed this trend, some of them show some divergence. “cancers”, “cardiovascular diseases”, “endocrine and metabolic diseases”, “other congenital disorders”, “reproductive system diseases”, “skin diseases”, and “other diseases” showed similarity to that of all disease-related proteins. On the other hand, “congenital disorders of metabolism”, “digestive system diseases”, “nervous system diseases”, and “urinary system diseases” showed another trend. The pProS-containing proteins in these categories do not have many CN proteins and N proteins but have proteins in other locations. It has been indicated that IDRs are found in large numbers in nuclear proteins, such as transcription factors and proteins in the signaling pathways [4,6]. The CN proteins and N proteins are those proteins under such categories because stimulus from outside of the cell is received by receptors and must be transmitted to the nucleus. The system therefore needs to have mobile proteins from cytoplasm to the nucleus. Additionally, in order to regulate such systems, information flow from the nucleus to the cytoplasm is required. Then, the pProS-containing proteins under the first trend can be the typical IDPs previously reported. The other locations contain “endoplasmic reticulum”, “cell projection”, and “cell junction”, etc. The proteins under the second trend suggest that IDPs not belonging to typical IDPs, such as transcription factors, can have functional sites in their IDRs and are involved in human diseases. For example, ARH as shown in Figure 3 has functions in the endocytosis in cytoplasm, and its IDRs have the key regions associated with hypercholesterolemia. 

## 4. Discussion

This study is based on IDR predictions and database annotations. We employed consensus methods to define IDRs using three predictors. Because the MobiDB-lite program contains eight different prediction models, we substantially used ten predictors to make consensus IDRs. We found experimental evidence for predicted IDRs in the detailed analysis of some proteins. For example, the annotations frequently found in the IDRs (Appendix A) were found in the regions experimentally verified as IDRs, and the examples of eIF4G1, SMN protein, and ARH also showed that some parts of the predicted IDRs have been experimentally verified to be IDRs (Figure 3). Due to these examples, we were convinced that the IDRs in this study were promising. Even with the conservative IDR definition, we found more than 1000 pProSs. If the condition to select IDRs was relaxed, more candidates for functional regions in IDRs could be obtained.

Functional segments on IDRs have been collected as the databases such as SLiM/ELM [72]. A proteome-wide analysis combined with high-throughput sequencing data showed that disease-related mutations were enriched in SLiMs on IDRs and occurred more frequently at functionally important residues in SLiMs [73]. Most of these motifs are annotated as “short sequence motif” in UniProt. In this sense, the analysis in this study convers not only these motifs but also potential functional segments annotated as “mutagenesis site” and “region of interest”. The ratio of the numbers of “short sequence motif” annotations to those of the other two is one-eighth (Table 1). This result suggests that knowledge of functional segments on IDRs can exist other than the information stored in the motif databases.

We searched potential binding segments in IDRs by referring to the UniProt annotations. The UniProt annotations are human-curated and highly reliable. Some of the information on a protein, however, does not appear in the FT section of the UniProt annotation. For example, the N–terminal IDR of p53 has 11 binding partners, and the C-terminal IDR has 15 partners solved in PDB structures (IDEAL: IID00015). The UniProt annotations in these IDRs are concise, as shown in Figure 2, although the links to the PDB entries are described in the “cross reference” section in UniProt. We picked eIF4G1 as an example in Figure 3. In this example, we found the pProS of the PABPC1-binding region in the UniProt annotation. However, this region also binds upon rotavirus nonstructural protein 3 [74], and this information does not appear in the feature table. In this sense, this study does not cover all of the knowledge of potential functional regions in IDRs. Thus, detailed analysis of each of the proteins would provide more information about functional segments on IDRs in the disease-related proteins.

In fact, we found potential functional sites from another information in UniProt. As shown in Figure 3, some of pProSs coincide with mutation sites associated with human disease. UniProt describes these associations as the link of mutation sites to the OMIM database, which are not directly linked with the feature annotations used in this study. Although we did not use such mutation site information to define pProS, we found considerable numbers of mutation sites in the predicted IDRs. We found 1611 mutation sites in the predicted IDRs of 572 proteins. Among them, UniProt describes links to the OMIM database for 919 mutation sites in 359 proteins. The list of these mutation sites can be found in Appendix A. Although it is not clear whether these mutation sites are binding sites for other molecules, these sites may possibly be regarded as functional regions in IDRs. Some of the readers may be interested in other model organisms. We briefly surveyed some of the represented model organisms, as shown in Appendix A. Although the results are preliminary, considerable numbers of annotations were found in the predicted IDRs for the mouse, rat, *Arabidopsis thaliana*, and yeast. These organisms have been long used for model organisms, and knowledge verified by experiments has been accumulated. Thus, for such model organisms, the strategy of the present study can be applied.

When we simply count the numbers of pProSs in the disease categories, the statistics were different from Table 2. The statistics are shown in Appendix A. The “pProS counts” represent how many times pProSs occur in each of the categories. “Cancers” were top ranked, followed by “congenital malformation” and “nervous system diseases”. The large numbers of proteins can account for the large numbers of “pProS counts” of “nervous system diseases” and “congenital malformation”. “Cancers”, however, does not have as many proteins as these two categories, in spite of many “pProS counts.” This is due to the multiple involvement of a protein in different diseases. When a protein having a pProS and is assigned to two diseases, we counted two for the pProS count. Thus, when a protein is assigned to multiple diseases in a disease category, redundant counts in the disease category occur. Appendix A lists the top 10 redundant proteins in terms of multiple disease annotations. p53 Has the maximum multiple disease annotations, and most of these diseases are in the category “cancers”. The multiple annotations on the other proteins tend to be also redundant in ”cancers”. The pProS redundancy in Appendix A, which is the ratio of the numbers of unique pProS to the numbers of pProS counts, shows the trend of this multiple disease association. “cancers” has a remarkably high value of pProS redundancy, suggesting that pProS-containing proteins in “cancers” tend to associate with several kinds of cancers.

p53 is involved in an extremely large number of diseases and has been also known as a hub protein that has great numbers of binding partners in PPI networks. In fact, the BioGRID database [75] lists 1056 interactors for p53, which is ranked 11th in terms of the number of PPIs in the human proteome. Then, we looked at the relation between the number of diseases assigned by KEGG DISEASE and the number of protein interactors. Figure 5 shows the relationship between the two. The correlation coefficients between the mean and median values of the numbers of interactors to the numbers of assigned diseases were 0.86 and 0.87. The distribution of the numbers of assigned diseases likely follows the power law. Therefore, most of the samples are found in the bins of the left side of the chart, and only small numbers of samples are found in the right side of the chart, as suggested by the long boxes. Even within the data of less than 10 assigned diseases, the correlation coefficient of the mean value is 0.73. It can therefore be said that there is generally a correlation between the numbers of interactors and the numbers of diseases involved. It has been pointed out that hub proteins in PPI networks are rich in IDRs [76,77]. If we accept the relationship between the number of protein interactors and the number of diseases involved, these hub proteins, namely IDPs, must be associated with many human diseases. 

Post-translational modifications (PTM) are an important modification on proteins to regulate their function. In particular, phosphorylation has been known to occur preferentially on IDRs [78] to regulate signaling pathways and other many biological processes. We surveyed phosphorylation sites of the pProS-containing proteins by using the UniProt annotations and found that 876 phosphorylation sites coincide with pProS regions in 290 proteins. Most of these phosphorylation related pProSs have the annotations relating to protein binding, suggesting that the PPI via these pProSs may be regulated by phosphorylation. We also checked links of these regions to the OMIM database and found that about 30 phosphorylation sites have the links to the OMIM database. Although the number of the direct link to OMIM is short, a lot of phosphorylation sites can regulate PPIs, and the defect of these phosphorylation related pProSs may make an impact on PPI networks.

We analyzed localizations of disease-related proteins to find the richness of the mobile proteins coming and going between the cytoplasm and nucleus in pProS-containing proteins. Recently, proteins and protein-nucleic acid mixtures can undergo liquid–liquid phase separation (LLPS) to form non-membrane organelles or lipid droplets, and have several important biological functions [79,80,81,82]. Intrinsically disordered regions have been shown to play important roles in LLPS [83,84,85]. Some of the non-membrane organelles appeared in the UniProt annotations of subcellular localization. Appendix A shows the distributions of these non-membrane organelles found in disease-related proteins. It is clear that the pProS-containing proteins are over-represented, while the non-pProS proteins are not, with the exception of “lipid droplet” and “P-body”. Although the numbers of annotations for these non-membrane organelles are small, these results suggest that IDPs play an important role in forming such organelles and have functional segments on their IDRs.

## 5. Conclusions

We conducted bioinformatics analyses to survey our knowledge on functional segments in IDRs from the perspective of disease-related proteins. We found more than a thousand annotations in the predicted IDRs, and considerable fractions of the disease-related proteins contained functional segments on IDRs. The detailed analysis on some of the examples showed that the pProSs found in this study were located in the experimentally verified IDRs and could directly associate with the diseases. Hub proteins in the PPI network tend to be involved in many human diseases, and some of the pProS-containing proteins are embedded in non-membrane organelles. We should note that the statistics in this study convey only current research, and more than 100,000 functional segments are expected to exist in IDRs. However, even with limited information, this study showed the power of database search for retrieving knowledge of functional segments in IDRs. The complete lists of pProSs can be found in Appendix A.

## Figures and Tables

**Figure 1 biomolecules-09-00088-f001:**
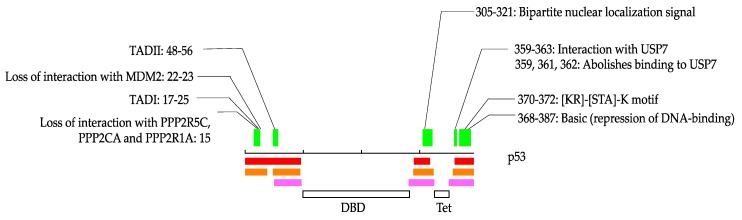
An example of possible protean segment (pProS) definition, illustrated by p53. The black line in the middle represents the amino acid chain, and the intrinsically disordered regions (IDR) predictions are presented below. Pink, orange, and red represent the results by MobiDB-lite, DISOPRED3, and DICHOT, respectively. Regions where any of the two methods predict IDR are defined as IDRs. The green bars represent pProSs, and the annotations defining pProS are shown above with residue numbers of the annotations. DBD: DNA-binding domain; Tet: Tetramerization domain.

**Figure 2 biomolecules-09-00088-f002:**
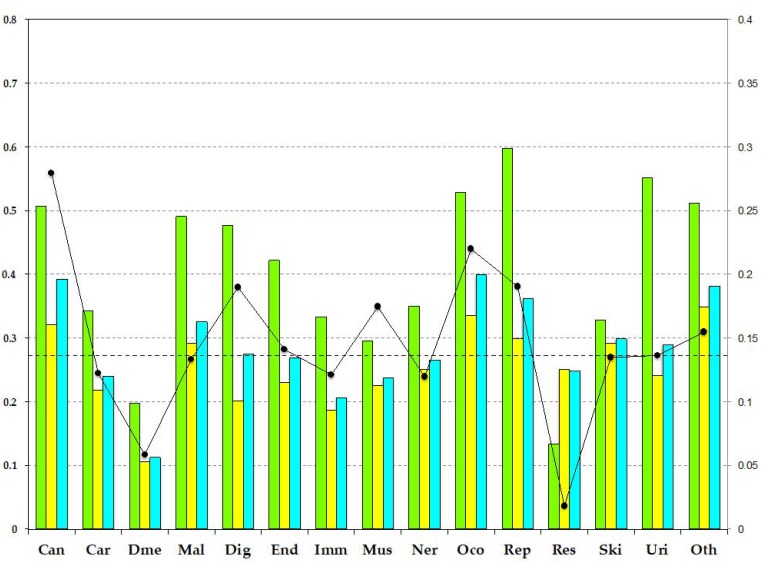
The IDR ratios by disease category. The green, yellow, and blue bars represent the IDR fractions of the pProS-containing proteins, the non-pProS proteins, and the total proteins in each of the disease categories, respectively. The measure on the left axis represents the IDR fractions. The black line with dots represents the protein coverage found in Table 2. The measure on the right axis represents the protein coverage. The dashed line represents the IDR ratio of the human proteome. Can: Cancers; Car: Cardiovascular diseases; Dme: Congenital disorders of metabolism; Mal: Congenital malformations; Dig: Digestive system diseases; End: Endocrine and metabolic diseases; Imm: Immune system diseases; Mus: Musculoskeletal diseases; Ner: Nervous system diseases; Oco: Other congenital disorders; Rep: Reproductive system diseases; Res: Respiratory diseases; Ski: Skin diseases; Uri: Urinary system diseases; Oth: Other diseases.

**Figure 3 biomolecules-09-00088-f003:**
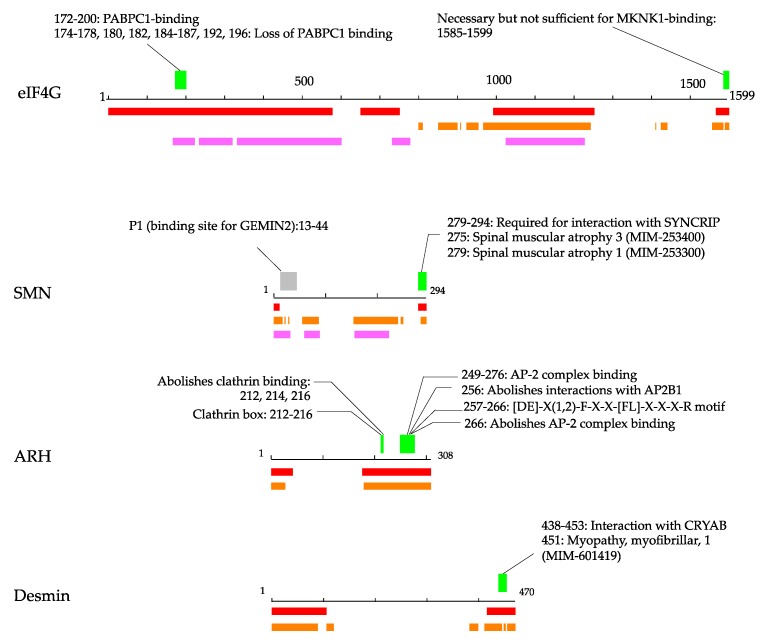
Examples of proteins with pProS. The black line in the middle represents the amino acid chain, and the IDR predictions are presented below. Pink, orange, and red represent the results by MobiDB-lite, DISOPRED3, and DICHOT, respectively. The green bars represent pProSs, and the annotations defining pProS are shown above with the residue numbers of the annotations. The gray bar in the example of survival of motor neuron (SMN) represents the regions of a pseudo-pProS, which was not taken as pProS because the region of the annotation is longer than 30. In the case of low-density lipoprotein receptor adaptor protein 1 (ARH) and desmin, MobiDB-lite does not predict any IDRs. The scale of eIF4G1 (eukaryotic translation initiation factor 4 gamma 1) differs from other three.

**Figure 4 biomolecules-09-00088-f004:**
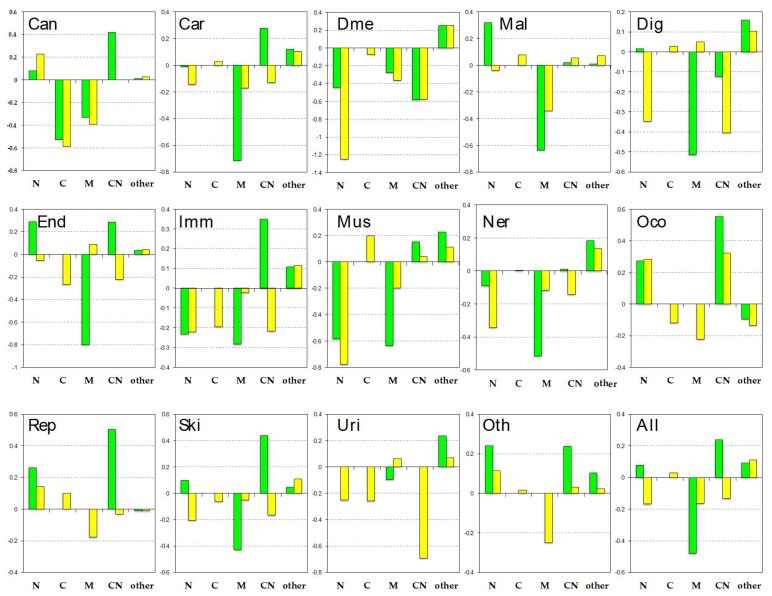
Subcellular localizations by disease category. The bars represent the degree of over-representation in each of the location categories, where green represents pProS-containing proteins, and yellow represents non-pProS proteins (see also Materials and Methods). N: Nuclear; C: Cytoplasm; M: Membrane; CN: Cytoplasm and nuclear; Can: Cancers; Car: Cardiovascular diseases; Dme: Congenital disorders of metabolism; Mal: Congenital malformations; Dig: Digestive system diseases; End: Endocrine and metabolic diseases; Imm: Immune system diseases; Mus: Musculoskeletal diseases; Ner: Nervous system diseases; Oco: Other congenital disorders; Rep: Reproductive system diseases; Ski: Skin diseases; Uri: Urinary system diseases; Oth: Other diseases; All: All disease-related proteins.

**Figure 5 biomolecules-09-00088-f005:**
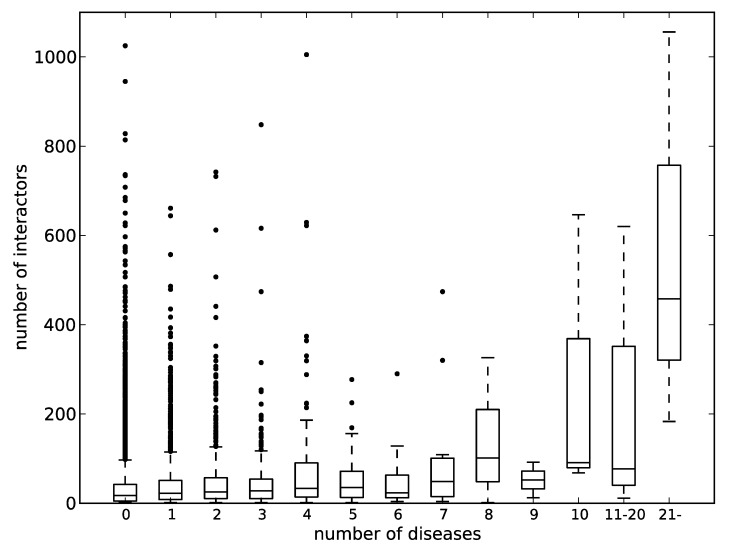
The correlation between the number of protein–protein interactions and the number of diseases involved. The horizontal axis represents the number of diseases, and the vertical one represents the number of interactors. A box and a pair of whiskers represent quartiles, and the line in the middle of the box represents the median. The dots represent outliers.

**Table 1 biomolecules-09-00088-t001:** Statistics of the UniProt annotations.

	All Proteins	Disease-Related	pProS	pProS (%)
No. proteins	20,410	3378	402	11.9
No. annotations shorter than 30 residues	29,145	18,450	1124	6.1
“Region of interest”	4646	2656	220	8.3
“Mutagenesis site”	21,269	14,056	479	3.4
“Short sequence motif”	3230	1740	425	24.4

pProS: Possible protean segment.

**Table 2 biomolecules-09-00088-t002:** Statistics of pProS by the disease category.

Category	No. Unique pProSs	No. Proteins with pProS	No. Proteins	Protein Coverage (%)	Average Annotations
Cancers	147	57	204	27.9	2.6
Cardiovascular diseases	93	41	335	12.2	2.3
Congenital disorders of metabolism	53	40	687	5.8	1.3
Congenital malformations	242	111	832	13.3	2.2
Digestive system diseases	32	15	79	19.0	2.1
Endocrine and metabolic diseases	63	30	213	14.1	2.1
Immune system diseases	56	31	256	12.1	1.8
Musculoskeletal diseases	69	26	149	17.4	2.7
Nervous system diseases	199	95	795	11.9	2.1
Other congenital disorders	39	20	91	22.0	2.0
Reproductive system diseases	21	12	63	19.0	1.8
Respiratory diseases	1	1	55	1.8	1.0
Skin diseases	22	14	104	13.5	1.6
Urinary system diseases	33	9	66	13.6	3.7
Other diseases	68	30	194	15.5	2.3

**Table 3 biomolecules-09-00088-t003:** The list of the proteins with long pProSs.

Protein Name	UniProt Accession	pProS Residues	No. Disease	Disease Category	Disease
DNA excision repair protein ERCC-6	Q03468	63	4	Ner	Age-related macular degeneration
				Ner	Cockayne syndrome
				Mal	Disorders of nucleotide excision repair
				Ski	Ultra violet-sensitive syndrome
Cellular tumor antigen p53	P04637	61	46		*
E3 ubiquitin-protein ligase RNF168	Q8IYW5	55	1	Imm	RIDDLE syndrome
CD2-associated protein	Q9Y5K6	50	1	Uri	Focal segmental glomerulosclerosis
Synaptic functional regulator FMR1	Q06787	47	3	Rep	Premature ovarian failure
Low-density lipoprotein receptor-related protein 2	P98164	44	1	Mal	Donnai–Barrow syndrome
Eukaryotic translation initiation factor 4 gamma 1	Q04637	44	1	Ner	Parkinson disease
DNA (cytosine-5)-methyltransferase 1	P26358	41	1	Ner	Hereditary sensory and autonomic neuropathy
Period circadian protein homolog 2	O15055	40	1	Ner	Familial advanced sleep phase syndrome
Latent-transforming growth factor β-binding protein 2	Q14767	40	1	Ner	Primary congenital glaucoma
Low-density lipoprotein receptor-related protein 6	O75581	39	3	Can	Breast cancer
				Car	Coronary artery disease
				Dig	Tooth agenesis
DNA damage-inducible transcript 3 protein	P35638	39	1	Can	Myxoid liposarcoma
KN motif and ankyrin repeat domain-containing protein 1	Q14678	39	1	Ner	Spastic quadriplegic cerebral palsy
Retinoic acid-induced protein 1	Q7Z5J4	36	1	Oco	Smith–Magenis syndrome
Histone-lysine *N*-methyltransferase 2D	O14686	35	2	Can	Follicular lymphoma
				Mal	Kabuki syndrome
FYN-binding protein 1	O15117	35	1	Car	Thrombocytopenia
Low-density lipoprotein receptor adapter protein 1	Q5SW96	33	1	Dme	Familial autosomal recessive hypercholesterolemia
Catenin β-1	P35222	32	8	Can	Thyroid cancer
				Can	Medulloblastoma
				Can	Endometrial cancer
				Can	Colorectal cancer
				Can	Gastric cancer
				Can	Hepatocellular carcinoma
				Oth	Autosomal dominant mental retardation
				Ski	Pilomatricoma
Sp110 nuclear body protein	Q9HB58	31	1	Dig	Hepatic veno-occlusive disease with immunodeficiency
Low-density lipoprotein receptor-related protein 5	O75197	30	6	Mal	Osteopetrosis
				Mal	Worth type autosomal dominant osteosclerosis
				Mal	Osteoporosis-pseudoglioma syndrome
				Mus	Hyperostosis corticalis generalisata
				Mus	Osteoporosis
				Ner	Familial exudative vitreoretinopathy
LEM domain-containing protein 2	Q8NC56	30	1	Ner	Cataract
Single-stranded DNA cytosine deaminase	Q9GZX7	30	1	Imm	Hyper IgM syndromes, autosomal recessive type

* The list of diseases involving p53 is found in Appendix A. Can: Cancers; Car: Cardiovascular diseases; Dme: Congenital disorders of metabolism; Mal: Congenital malformations; Dig: Digestive system diseases; End: Endocrine and metabolic diseases; Imm: Immune system diseases; Mus: Musculoskeletal diseases; Ner: Nervous system diseases; Oco: Other congenital disorders; Rep: Reproductive system diseases; Res: Respiratory diseases; Ski: Skin diseases; Uri: Urinary system diseases; Oth: Other diseases.

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
