# Peer review of "Functional Segments on Intrinsically Disordered Regions in Disease-Related Proteins"

_biomolecules, 2019, doi:10.3390/biom9030088_

Round 1
Reviewer 1 Report
It's an interesting study on the part of the authors. I have few comments/suggestions, though:
1) Authors should consider a detailed comparison of their study with previous studies performed in the field on IDRs and clearly decipt how their study is a major contribution?
2) I would also encourage authors to consider other important species data like mouse and yeast to include in their present study.
3) The computational tools authors used to obtain the results should be described in more detail so a reader can understand and reproduce results.
4) The IDRs are also a hug for posttranslational-modifications especially phosphorylation, I would encourage authors to consider analyzing PTMs present in IDRs and also variations impact PPI and PTMS present in IDRs.
Author Response
Thank you for your review on our manuscript entitled "Functional segments on intrinsically disordered regions in disease-related proteins". We revised the manuscript, and listed the answers for the comments of the reviewers.
For the reviewer 1,
Comment: 1) Authors should consider a detailed comparison of their study with previous studies performed in the field on IDRs and clearly decipt how their study is a major contribution?
Reply: We added some sentences to describe comparisons with some of the previous studies. They can be found in the lines from 221 to 224, from 237 to 241, and 376 to 384.
Comment: 2) I would also encourage authors to consider other important species data like mouse and yeast to include in their present study.
Reply: It is too short time to conduct exactly the same analysis on other model organisms, because the deadline of the revise is 5 days. We, however, have preliminary results of some model organisms. Although the condition of the analysis is not exactly the same with the present analysis, we think that it is worth being included in the manuscript to know the possibility of the application to other organisms. Then, we added some sentences in the lines from 406 to 411 and a new table, Table S4.
Comment: 3) The computational tools authors used to obtain the results should be described in more detail so a reader can understand and reproduce results.
Reply: We added some sentences to describe the details of each of the methods in the lines of 81, from 85 to 88, from 89 to 92, and from 95 to 97. We also added a new figure, Figure S1, to describe the brief outline of the procedure.
Comment: 4) The IDRs are also a hug for posttranslational-modifications especially phosphorylation, I would encourage authors to consider analyzing PTMs present in IDRs and also variations impact PPI and PTMS present in IDRs.
Reply: We surveyed the coincidence of phosphorylation sites with the pProSs by using the UniPort annotations. We discussed the results from the aspect of PPI networks to add some sentences in the lines from 447 to 457.
Reviewer 2 Report
In this manuscript, the authors have conducted a survey of functional segments in IDRs and found >1000 potential functional IDR segments in disease‐related proteins. This is an interesting study. I would recommend including a figure outlining the methodology.
Table 2 provides the statistics of pProS by the disease category. Please include a supplementary table with the names, IDs, related diseases and other relevant information for all the proteins in table 2.
Please add axis labels to the graphs
Author Response
Thank you for your review on our manuscript entitled "Functional segments on intrinsically disordered regions in disease-related proteins". We revised the manuscript, and listed the answers for the comments of the reviewers.
For the reviewer 2,
Comment: In this manuscript, the authors have conducted a survey of functional segments in IDRs and found >1000 potential functional IDR segments in disease‐related proteins. This is an interesting study. I would recommend including a figure outlining the methodology.
Reply: We added a new figure, Figure S1, to describe the brief outline of the procedure.
Comment: Table 2 provides the statistics of pProS by the disease category. Please include a supplementary table with the names, IDs, related diseases and other relevant information for all the proteins in table 2.
Reply: We added disease category name, disease sub-category name, and disease name to Table S7 (previous Table S6). Now, Table S7 has uniprot accession, protein name, pProS region, IDR region, disease category, disease sub-category, and disease name for all of the pProS-containing proteins. Although this table contains all of the pProS containing proteins not limiting the disease-related proteins, we think that this file is enough to know pProSs in the disease-related proteins by using Excel function.
Round 2
Reviewer 1 Report
Authors accepted the suggestions and further improved manuscript. Congratulations.
I strongly suggest re-editing paper after going through thoroughly, language needs to be improved to do justice with the work and the readers.